# OpenReview forum: "TokSuite: Measuring the Impact of Tokenizer Choice on Language Model Behavior"
_ICLR.cc/2026/Conference — Submitted to ICLR 2026_

### Official Review · Reviewer_7Sj1 · 2025-10-26

**Soundness:** 3
**Presentation:** 2
**Contribution:** 3
**Rating:** 4
**Confidence:** 4

**Summary:**

The paper introduces TokSuite, a suite of LLMs to study the effect of tokenizer choice on LLMs's behavior and robustness. In addition to the models the paper also introduces several benchmarks to study the robustness of these models under varying real-world like perturbations.

Some key results are -

- Different tokenizers perform in different ways on different perturbations (TokenMonster which is English-only is more robust to some perturbations as compared to multilingual tokenizers)
- Byte tokenizers are more robust to multilingual perturbations and subword fragmentation.
- Tokenizer significantly impacts processing ability of certain types of content such as technical content like LaTeX content due to lossy pre-processing

**Strengths:**

- The paper is clearly motivated and tests several different off-shelf popularly used tokenizers
- The results on the author created benchmarks show clear effects of tokenizer changes
- The unification of shared token initialization is a nice step and shows attention to detail
- Technical content analyses section is quite interesting as well as relevant

**Weaknesses:**

- I feel an interesting and important axes is comparing the effect of vocab size within a single tokenization method. For example if over the same corpus and using the same tokenization/pretokenization algorithm, the authors learnt multiple vocabs of varying sizes and saw changes in performance then that would be a more clear indicator of effect of tokenizer vocab size as at the moment comparing across tokenizers does not make a lot of sense given a several times larger tokenizer leads to many more parameters for the model to learn.
- The claim in Lines 84-85 seems to be unsupported as it could be because tokenizers are undertrained for non-English languages and had you learnt the vocab over a corpus which has several of the commonly occurring real world issues with other languages the tokenizer would have tokens to handle those issues more gracefully
- The findings about byte-level models seems fairly obvious to me given that is their entire selling point?
- It is hard to make sense of numbers like how different is 0.04 v/s 0.06 drop in line 359. Does it even matter or is it a big difference?
- The custom benchmarks seem quite small and differences may be in bound of variance?

**Questions:**

- I'm assuming data order is also same? (Line 54-55)
- Line 197, don't you mean intersection?

---

> ### Author Response · Authors · 2025-11-21
>
> Thank you for your thoughtful review. Please find our replies below for the quoted points.
>
>
> ## Effect of vocab size
> > I feel an interesting and important axes is comparing the effect of vocab size within a single tokenization method.
>
>
> We agree that the effect of vocabulary size is an interesting dimension for tokenizer analysis. However, in this work we focus on off-the-shelf tokenizers which are in use by the community and preexisting tokenizers are generally not released with a range of vocabulary sizes. Previous works have explored the effect of vocabulary size (https://openreview.net/pdf?id=ZFYBnLljtT, https://arxiv.org/abs/2407.13623, https://arxiv.org/pdf/2402.18376, https://arxiv.org/pdf/2406.16508), but they had to train their tokenizers from scratch.
>
> ## Findings
> > The claim in Lines 84-85 seems to be unsupported as it could be because tokenizers are undertrained for non-English languages and had you learnt the vocab over a corpus which has several of the commonly occurring real world issues with other languages the tokenizer would have tokens to handle those issues more gracefully
>
> Thank you for highlighting this nuance. TokSuite includes a wide-range of tokenizers, at least half of which has large vocabulary and were trained in non-English data (See Table 5 for fertility and parity as a signal of coverage). The models that are more efficiently compressing non-English data are not necessarily handling non-English subsets best, while TokenMonster, which was trained only on English, provides an interesting contrast.
>
> > The findings about byte-level models seems fairly obvious to me given that is their entire selling point?
>
> We agree that some of our results have been hinted at in prior work but our work is the first to compare these tokenizers in a truly controlled setting and tokenization-specific benchmark.
>
> > It is hard to make sense of numbers like how different is 0.04 v/s 0.06 drop in line 359. Does it even matter or is it a big difference?
>
> We report the average performance degradation, i.e., how much of the canonical performance is preserved on the perturbed examples. So a 0.04 degradation means that there is a 4% performance drop. We agree that this makes the relative comparison across categories hard to interpret, to address this we have 1) created a new view of the main table (Table 1) with a heatmap highlighting the best and worst models across each category in Table 22. We have normalized the scores in each column (x - min_of_col) / (max_of_col - min_of_col). 2) Added standard deviations across 10,000 bootstrap trials in Table 21.

---

> > ### Author Response · Authors · 2025-11-21
> >
> > ## Scale of the benchmark
> > > The custom benchmarks seem quite small and differences may be in bound of variance?
> >
> > We agree that the benchmark could be expanded (also raised by 7UAN) but believe that for the first version of a benchmark that is hand-curated and validated by native speakers, it represents a strong and necessary starting point. Unlike work focusing on randomly generated perturbations, we specifically target real-world scenarios that naturally occur in a multilingual environment.
> > To this end, the four languages in the study cover different linguistic intricacies :
> > - Farsi uses the Arabic script and presents unique challenges where the same text can be represented by optional diacritics, which break tokenization.
> > - Mandarin Chinese not only uses a different script but is also an isolating language. We also cover Pinyin and its errors, which is rarely found in the training data but is an essential part of daily communication.
> > - Turkish, an agglutinative language with six additional letters in its alphabet, is rich in grammar that severely impacts word form and tokenization.
> > - Italian is representative of fusional Latin languages with complex inflectional patterns and accents.
> >
> > TokSuite includes 40 parallel questions in all five languages that could further be used in studies investigating multilingual knowledge. We select language-specific as well as common perturbations so that the perturbation could be applied to most---all in many cases---canonical questions in that language. Each language has >485 examples (Table 5), which is significant for a hand-curated dataset. We hope that the failure modes observed in our study, and the fact that no single model can handle all categories, serves as a critical first step for the community to pay closer attention to these multilingual robustness issues.
> >
> > As for the statistical significance of our results (also raised by p3g4), we estimate distributional statistics through a 10,000-trial bootstrap, producing standard deviations that have been added to our main results table (Table 21 in the Appendix). We highlight that all of our discussed findings deal with performance differences that are larger than one standard deviation.
> >
> > ## Experimental Details
> > > I'm assuming data order is also same? (Line 54-55)
> >
> > This is an important methodological point, and we appreciate the chance for clarification (also raised by reviewer p3g4). The dataloader we use is deterministic, i.e. it samples documents in the same order. However, the tokenizer differences result in variation in the tokenized batch streams, which results in slightly different batch ordering.
> >
> > > Line 197, don't you mean intersection?
> >
> > No, we specifically compute the union of all vocabularies. The shared initialization for subword tokens that are present in multiple tokenizers could have been implemented from an intersection of vocabs and a multi-step model parameter initialization procedure. However, it was simpler to take the union as it let us implement initialization with a single indexing operation.

---

> > > ### Comment · Reviewer_7Sj1 · 2025-11-24
> > >
> > > Thanks for the clarifications
> > > I've updated my score

---

### Official Review · Reviewer_RRpy · 2025-10-29

**Soundness:** 3
**Presentation:** 2
**Contribution:** 2
**Rating:** 4
**Confidence:** 3

**Summary:**

This paper aims to systematically explore the robustness of different tokenizers and their impact on language model performance, attempting to isolate the effects of the tokenizer from other training factors.
In addition, the authors propose the TokSuite Benchmark, designed to evaluate the robustness of various tokenizers.

**Strengths:**

1.The authors constructed a high-quality TokSuite Benchmark.

2.The authors evaluated 14 different tokenizers and provided several insightful conclusions based on the experimental results.

**Weaknesses:**

1.In Table 1, the average (Avg.) results across the 14 tokenizers show little variation—except for TokenMonster and Tekken, the remaining 12 tokenizers perform similarly. This raises the question of whether the benchmark is sufficiently sensitive to differentiate between the effectiveness of different tokenizers.

2.The benchmark does not evaluate generation, translation, or code-related tasks. Since the impact of tokenization can vary significantly across task types, the applicability of the conclusions may be limited.

3.Regarding the writing and presentation, the main text is text-heavy. Apart from a single table displaying the experimental results, there is a lack of additional visual aids—such as figures or more comprehensive tables—to help readers better understand the findings.

**Questions:**

1.In Section 3.3, the authors mention that “training models with different tokenizers under the same token budget means that each model has seen a different collection of text.” Given that the actual text exposed to each model can vary significantly due to tokenizer differences, how might this strategy of aligning only the token budget, but not the text, affect the fairness of tokenizer evaluation?

2.The authors choose LLaMA-1B as the model for evaluation. Could they briefly explain the rationale behind selecting this model? Moreover, since the vocabulary size of a tokenizer is often considered to be most effective when adapted to the model’s size, does using a fixed-size model (LLaMA-1B) for all tokenizers—regardless of their vocabulary scale—potentially suppress the performance of tokenizers that are designed for larger or smaller models?

---

> ### Author Response · Authors · 2025-11-21
>
> Thank you for your insightful review. Please find our replies below for the quoted points.
> ## Benchmark
> > 2.The benchmark does not evaluate generation, translation, or code-related tasks. Since the impact of tokenization can vary significantly across task types, the applicability of the conclusions may be limited.
>
> We agree that the tokenizer's impact may vary across different task types and these domains include many interesting examples. Our current framework is specifically designed to measure the intrinsic robustness of base models at the 1B parameter scale, which informed our task selection:
> - Generation (Proxy): Our multiple-choice and completion-style questions serve as a measurable proxy for generation that is amenable to evaluation on base models. Relying on fully open-ended generation tasks, especially in a multilingual context, often introduces significant noise and reliability issues into the evaluation process. Our format allows for clear, reproducible measurement of tokenization effects.
> - Translation and Code (Too Weak, Empirically Validated): We intentionally excluded these tasks because base models are typically too weak to reliably perform complex translation or code generation. We validated this by training a proxy model on a combined corpus (EN, TR, FA, ZH, IT, and a subset of the Stack-edu dataset), and its performance was found to be unreliable on standard coding benchmarks. Including such tasks would likely obscure the tokenization effects we aim to study at this capacity.
>
> ## Presentation
>
> > 3.Regarding the writing and presentation, the main text is text-heavy. Apart from a single table displaying the experimental results, there is a lack of additional visual aids---such as figures or more comprehensive tables---to help readers better understand the findings.
>
> We appreciate this feedback and fully agree. We have added a new figure (Figure 1, left) that explicitly shows the differences in sub-word tokenization between canonical and perturbed versions for the word “doctor” from the benchmark, which directly illustrates the core mechanism of failure. We have also added a figure summarizing the key features of the training (Figure 1, right). We have also added a heatmap version (Table 22) of the main table (Table 1) to highlight the models that experience the least (blue) and largest (red) performance degradation across categories.
>
> As our paper pertains to the critical difference in tokenization of natural text, we would also appreciate any further suggestion from the reviewer on how to improve the presentation and visual clarity of these underlying mechanisms.

---

> > ### Author Response · Authors · 2025-11-21
> >
> > ## Training Details
> > > 1.In Section 3.3, the authors mention that “training models with different tokenizers under the same token budget means that each model has seen a different collection of text.” Given that the actual text exposed to each model can vary significantly due to tokenizer differences, how might this strategy of aligning only the token budget, but not the text, affect the fairness of tokenizer evaluation?
> >
> > This is an important methodological point, and we appreciate the chance for clarification (also raised by reviewers 7Sj1 and p3g4). The dataloader we use is deterministic, i.e. it samples documents in the same order. However, the tokenizer differences result in variation in the tokenized batch streams, which results in slightly different batch ordering and different number of actual bytes consumed.
> >
> > While each model has indeed consumed different amounts of data, this is an inherent consequence of a tokenizer’s compression ability, which directly dictates how much raw information (in bytes/documents) is consumed for a fixed token budget, and this is the current practice in LLM training and reporting.
> >
> >
> > To investigate this tradeoff directly, we first reconstructed the entirety of the text data consumed by each model. To do so, we reran our deterministic data pipeline, detokenized each batch (datasets to be released) and computed the total number of UTF-8 bytes seen during training for each tokenizer. ByT5 has consumed 100GB of text, while TokenMonster, GPT-2, Comma, and Phi-3 are in the 215 to 287GB range, Llama around 397GB, and rest (Qwen3, Tekken, BLOOM, mBERT, GPT-4o, Aya, Gemma-2, and XGLM) between 411 to 477GB of text. Detailed results have been added to the appendix. Crucially, the models that saw more data were not necessarily the best!
> >
> >
> > > 2.The authors choose LLaMA-1B as the model for evaluation. Could they briefly explain the rationale behind selecting this model?
> >
> > We chose the LLaMA-1B architecture primarily for two reasons: standardization and computational tractability. Llama 1B is a reasonable choice to include modern, best-practice architectural details and training.
> >
> > Parameter size: We chose the 1B scale since we have a limited computational budget, but consider this scale to produce realistic and meaningful results.  There have been many performant models at this scale, including Qwen-3 0.6B and 1.7B, Llama-3.2 1B and 3B, Gemma-3 1B, and the 270M!, 3B SmolLM models.  Our choice of this scale is also practical – we lack the computational budget to train larger-scale models, especially in the large quantity we consider in our paper.
> >
> >
> > > Moreover, since the vocabulary size of a tokenizer is often considered to be most effective when adapted to the model’s size, does using a fixed-size model (LLaMA-1B) for all tokenizers---regardless of their vocabulary scale---potentially suppress the performance of tokenizers that are designed for larger or smaller models?
> >
> >
> > The core goal of our study is to isolate the effect of the tokenizer from all other variables. To achieve this, it is essential that the model architecture, parameter count, and training regimen are held identical across all tokenizer variants. While this could be a confounding factor across different scales,our choice is in line with the current practice where models at different scales are being trained with the same tokenizer within the same architectural family.
> >
> >
> > While Tao, et al. 2024 (2407.13623) find that the vocabulary size should be scaled with respect to the parameter size, the common practice is not in line with this with model families often sharing the same tokenizer across different parameter scales.
> >
> >
> > As a proxy for the effect of vocabulary size and model size, we evaluated Qwen-3 family (with parameters ranging from 0.6B to 14B for dense models and a 30B MOE). Results are provided in Table 20. Note that according to their technical report are all these models were pre-trained for 36T tokens (Qwen Team, 2025. Section 3.1) with the same tokenizer and highly likely on the same corpus (?). We see that the average robustness across all categories is in a close range for all models, albeit with non-systematic differences in individual categories.

---

> > > ### Comment · Reviewer_RRpy · 2025-11-27
> > >
> > > Thank you for the authors’ responses and for the additional clarifications. However, several concerns remain unaddressed.
> > >
> > > First, regarding the earlier question about the results in Table 1, the authors did not provide any direct response.
> > >
> > > Second, with respect to the reply to “Training Details 2,” the newly added experiments in Table 20 raise additional concerns. According to the results, the Qwen 1.7B model has an average robustness of 0.17, whereas the 8B model has 0.12, yielding a difference of 0.05. In contrast, in the main paper’s Table 1, the largest Avg gap across the 14 tokenizers is 0.10, and the difference between the second-highest and second-lowest Avg is only 0.04 < 0.05. This cross-scale comparison does not seem to support the authors’ claim that “the average robustness across all categories is in a close range for all models,”
> > >
> > > This discrepancy may further call into question the justification for selecting the LLaMA-1B scale as the primary experimental setting. It may also suggest that the proposed benchmark exhibits limited discriminative power across the 14 tokenizers.
> > >
> > > Finally, in the response to Benchmark Question 2, the authors mention that the exclusion of Translation and Code tasks is empirically validated. Could the authors please indicate where the corresponding experimental results have been added?

---

> > > > ### Author Response · Authors · 2025-12-03
> > > >
> > > > Thank you for your additional comments.
> > > >
> > > >
> > > > ## Benchmark
> > > > > First, regarding the earlier question about the results in Table 1, the authors did not provide any direct response.
> > > >
> > > > We apologize for forgetting to include our response to this question in our initial reply and appreciate the opportunity to clarify our position (a point also raised by reviewers p3g4 and 7Sj1). We agree that the aggregate average performance is similar; however, TokSuite reveals distinct and statistically significant failure modes across specific perturbation categories:
> > > >
> > > > **Statistical significance:** To ensure statistical confidence, we estimate distributional statistics through a 10,000-trial bootstrap, producing standard deviations that have been added to our main results table (Table 21 in the Appendix as well as the visualization in Figure 6 in the Appendix). We further perform Paired Wilcoxon Signed-Rank Test to confirm that every category (except for Unicode) has at least one significantly more robust and one significantly less robust model (See Table 22 in the Appendix). We revised the manuscript and highlight that **all of our discussed findings deal with performance differences that are larger than one standard deviation.**
> > > >
> > > > **Presentation:** Newly added Figure 6 and our analysis in the findings section reveal non-trivial strengths and shortcomings of current tokenizers. The benchmark is sensitive because it demonstrates that **no single tokenizer excels universally**. The same conclusion applies to vocabulary size (Figure 7) and underlying algorithm (Figure 8).
> > > >
> > > > We emphasize that the perturbations in TokSuite are simple, realistic variations of text that **a native speaker would identify with perfect accuracy**. Ideally a good multilingual multi-purpose tokenizer/model pairing should be able to differentiate them too. The persistent inability of modern tokenizers to handle all categories highlights a fundamental robustness weakness that our benchmark successfully identifies.
> > > >
> > > > ## Additional tasks
> > > > > Finally, in the response to Benchmark Question 2, the authors mention that the exclusion of Translation and Code tasks is empirically validated. Could the authors please indicate where the corresponding experimental results have been added?
> > > >
> > > > We apologize for the confusion. This experiment was part of the initial design phase and informed our task choice rather than being included as a result.
> > > >
> > > > In the initial phase, we trained a model with the Gemma-2 tokenizer on a corpus of English, Turkish, Farsi, Italian, Chinese, and StackEdu-v2 (with mixing ratios of 0.5-0.1-0.1-0.1-0.1-0.1). We observed noisy and unreliable signals on HumanEval, a coding benchmark. We subsequently reallocated the training budget to focus exclusively on high-quality natural language text.
> > > >
> > > > While we agree that the coding domain is a natural expansion of our benchmark with interesting cases relevant for tokenization, the models and training budget required to have meaningful performance on coding goes beyond the scope of our paper.

---

> ### Author Response · Authors · 2025-12-03
>
> ## Scale
>
> > Second, with respect to the reply to “Training Details 2,” the newly added experiments in Table 20 raise additional concerns. According to the results, the Qwen 1.7B model has an average robustness of 0.17, whereas the 8B model has 0.12, yielding a difference of 0.05. In contrast, in the main paper’s Table 1, the largest Avg gap across the 14 tokenizers is 0.10, and the difference between the second-highest and second-lowest Avg is only 0.04 < 0.05. This cross-scale comparison does not seem to support the authors’ claim that “the average robustness across all categories is in a close range for all models,”
>
> We apologize for the confusion caused by relying on the Qwen-3 scaling results. Please note that not all the Qwen-3 models were trained with the same recipe. Referring to the Qwen 3 [Technical Report](https://arxiv.org/pdf/2505.09388), on page 4 it is reported that “At the first pre-training stage, all Qwen3 models are trained on over 30 trillion tokens using a sequence length of 4,096 tokens.”  the authors also mention distillation for smaller models: “For smaller models, we use strong-to-weak distillation, leveraging both off-policy and on-policy knowledge transfer from larger models to enhance their capabilities.” on page 2. This distillation process introduces a significant training difference that makes the Qwen scaling results unreliable for drawing controlled conclusions about tokenizer robustness across scales.
>
> **New Data/Experiment** To address these concerns directly, we trained a **7B model** with the Llama-3.2 tokenizer using the same exact training recipe and data as the 1B model. This means the larger model has seen the exact same raw text in the exact same order as the 1B Llama-3.2 model, differing only in parameter scale.
>
> The results from this controlled experiment confirm our core claim throughout the rebuttal period: scaling does not translate to significant gains in robustness and the 1B parameter scale remains a tractable and suitable scale to study tokenization robustness. While the canonical performance on the benchmark does improve with scale (see Table X.2), it performs roughly the same in all categories except those related to noise (Table X.1).
>
> Table X.1: Scaling experiment - Comparing 7B and 1B model that is trained with identical tokenizer and training recipe (We added this table to the manuscript as Table 20)
>
> | Model              |   Input (Non-EN) |   Diacritics (Non-EN) |   Orthographic Errors (EN) |   Orthographic Errors (Non-EN) |   Morphological (EN) |   Morphological (Non-EN) |   Noise (EN) |   Noise (Non-EN) |   LaTeX |   STEM (EN) |   Unicode |   Average |
> |:------------------------|-----------------:|----------------------:|---------------------------:|-------------------------------:|---------------------:|-------------------------:|-------------:|-----------------:|--------:|------------:|----------:|----------:|
> | 7B|0.3  |0.52 |0.05 |0.06 |0.24 |0.08 |0.09 |0.14 |    0.17 |0.26 |      0.6  |      0.23 |
> |1B|0.33 |0.55 |0.11 |0.1  |0.25 |0.08 |0.15 |0.24 |    0.18 |0.29 |0.59 |      0.26 |
>
>
> 7B model does perform better in the canonical subset but not in the perturbed subsets.
>
> Table X.2: Canonical performance improves with scale
>
> | Model              |   Chinese Canonical |   English Canonical |   Farsi Canonical |   Italian Canonical |   Math Canonical |   STEM Canonical |   Turkish Canonical |   Average |
> |:------------------------|----------------------------------------------------:|----------------------------------------------------:|--------------------------------------------------:|----------------------------------------------------:|-------------------------------------------------:|-------------------------------------------------:|----------------------------------------------------:|----------:|
> | Llama-3.2-7B |0.9   |1|0.83 |0.98 |1|0.93 |0.93 |  0.94 |
> | Qwen-3|0.7   |0.95  |0.85  |0.98 |0.95 |0.89 |0.85  |  0.88 |
> | Llama-3.2|0.8   |1|0.75  |0.98 |0.81 |0.89 |0.85  |  0.87  |
> | XGLM|0.88 |0.95  |0.8   |0.93 |0.71 |0.89 |0.85  |  0.86 |
> | Gemma-2|0.78 |0.98 |0.75  |0.98 |0.81 |0.89 |0.83 |  0.86 |

---

### Official Review · Reviewer_p3g4 · 2025-10-31

**Soundness:** 3
**Presentation:** 3
**Contribution:** 3
**Rating:** 6
**Confidence:** 3

**Summary:**

This paper introduces TokSuite to explore the impact of tokenizer choice on LM behavior, addressing a critical confounding variable in model evaluation. The authors train 14 identical 1B-parameter models, varying only the tokenizer, using a novel "super vocabulary" for shared initialization. To evaluate them, TokSuite also provides a new 5-language benchmark focused on robustness to real-world perturbations like typos and Unicode styling. The study finds that tokenizer algorithms can be more critical for robustness than vocabulary size. The work also identifies universal failures across all models when handling STEM notation and Unicode formatting.

**Strengths:**

* This paper conducts an in-depth study of tokenizers in large language models. By keeping all other conditions constant and varying only the tokenizer, the authors train 14 different models and perform a unified comparison.
* The design of the TokSuite benchmark is highly targeted. Instead of using standard, clean evaluations, it focuses on “perturbations” that specifically test tokenizer weaknesses, with particular attention to multilingual, math/STEM, and Unicode formatting scenarios.
* The paper carries out extensive and detailed experiments, revealing many new insights and findings.

**Weaknesses:**

* Most of the reported results only include the mean values, lacking variance and statistical significance tests. Given that many tokenizers show only small differences in performance, could these results be influenced by random factors?
* In this paper, the authors control the total number of training tokens to be consistent across models, but the total number of training texts varies significantly between datasets. Could this be a major factor contributing to performance differences, such as disparities in language knowledge? Therefore, I believe the authors should include experiments where the total number of training texts is also controlled to ensure greater completeness and fairness of the study.

**Questions:**

* Could the authors provide further scaling experiments on tokenizers, including variations in model size and data volume? I’m curious whether the characteristics of these tokenizers would change when scaled up.
* Could the authors include experiments where the total number of training texts is controlled to be consistent? This would make it easier to compare the impact of different tokenizers under equivalent knowledge conditions.
* Would it be possible to first train on a superset tokenizer, and then continue pretraining on each specific tokenizer? This could help ensure that the number of texts seen does not differ too much, thereby reducing knowledge discrepancies.
* Could the authors provide more detailed experimental results, including variance and statistical significance tests?

---

> ### Author Response · Authors · 2025-11-21
>
> Thank you for your substantive review. Please find our replies below for the quoted points.
>
> ## Statistical Significance
> > Most of the reported results only include the mean values, lacking variance and statistical significance tests.
>
> To address this concern (also raised by reviewer 7Sj1), we estimate distributional statistics through a 10,000-trial bootstrap, producing standard deviations that have been added to our main results table (Table 21 in the Appendix). We highlight that all of our discussed findings deal with performance differences that are larger than one standard deviation.
>
>
> ## Scale and Experimental Setting
> > In this paper, the authors control the total number of training tokens to be consistent across models, but the total number of training texts varies significantly between datasets. Could this be a major factor contributing to performance differences, such as disparities in language knowledge? Therefore, I believe the authors should include experiments where the total number of training texts is also controlled to ensure greater completeness and fairness of the study.
>
>
> This is an excellent point and was part of our initial design considerations.
>
> While each model has indeed consumed different amounts of data, this is an inherent consequence of a tokenizer’s compression ability, which directly dictates how much raw information (in bytes/documents) is consumed for a fixed token budget, and this is the current practice in LLM training and reporting.
>
> To investigate this tradeoff directly, we first reconstructed the entirety of the text data consumed by each model. To do so, we reran our deterministic data pipeline, detokenized each batch (datasets to be released) and computed the total number of UTF-8 bytes seen during training for each tokenizer. ByT5 has consumed 100GB of text, while TokenMonster, GPT-2, Comma, and Phi-3 are in the 215 to 287GB range, Llama around 397GB, and rest (Qwen3, Tekken, BLOOM, mBERT, GPT-4o, Aya, Gemma-2, and XGLM) between 411 to 477GB of text. Detailed results and a short discussion have been added to the appendix in Table 4. Crucially, the models that saw more data were not necessarily the best!
>
> > Could the authors provide further scaling experiments on tokenizers, including variations in model size and data volume? I’m curious whether the characteristics of these tokenizers would change when scaled up.
>
> While we are unable to perform a comprehensive study with a larger architecture, we could use the different scale models within an architectural family, e.g. Qwen-3 0.6B to 30B, as a proxy to answer this question, which share the same tokenizer, see Table 20.
>
> > Could the authors include experiments where the total number of training texts is controlled to be consistent? This would make it easier to compare the impact of different tokenizers under equivalent knowledge conditions.
>
> We appreciate this suggestion and agree that it is an interesting ablation. We are currently performing a follow-up experiment where three models are trained on the same quantity of UTF-8 text bytes (rather than the same quantity of tokens). We have started training three new models based on the Llama, Gemma-2, and Phi-3 tokenizers, varying the training steps so that each model sees the exact same text that the Llama tokenizer-based model was trained on, allowing for a direct comparison of tokenizer impact under equivalent knowledge conditions. We will update our response and submission when these training runs complete.
>
> > Would it be possible to first train on a superset tokenizer, and then continue pretraining on each specific tokenizer? This could help ensure that the number of texts seen does not differ too much, thereby reducing knowledge discrepancies.
>
> We agree that this is an interesting idea! Unfortunately, it is beyond the scope of TokSuite. Our super-vocabulary contains 942,440 items, which would make the embedding table huge (~4B parameters)!

---

> > ### Comment · Reviewer_p3g4 · 2025-11-28
> >
> > Thank you for your efforts! I will keep my positive scores.

---

### Official Review · Reviewer_7UAN · 2025-11-02

**Soundness:** 2
**Presentation:** 2
**Contribution:** 2
**Rating:** 4
**Confidence:** 4

**Summary:**

The propose a set of tests to understand the impact of tokenizer choices and train a set of lms to evaluate on the benchmark.

Note that my scores will be set after discussion. So dont take my scores serious please.

**Strengths:**

The study is systematic and broad.

**Weaknesses:**

Training new models has the problem that the models are comparetively small and hence one needs to wonder if the findings are transferable

I would have loved to see the byte latent transformer in the study as well as decoding methods that target byte level decoding such as Phan 2025.

**Questions:**

Generally i think this is a wonderful study here are a few things i would like clarification on


Can you justify the choices of the benchmark for me? Do you know if this is in any sense complete?

I also wonder about langruages like chinese or thai. During generation one token may span only part of the semantic meaning aka the opposite of the prompt boundary problem. Is there a chnace to get an evaluation in that adresses that.

In appendix a can you clarify how all the different bpe tokenizers differ? They all use the same code to my understanding but with different settings what are the settings?


To summarize my biggest concern. You will have to clarify the scope and hence value of the study a bit better. I think the mean reader will say well 5is does not generalize why would i trust the conclusions for my work.

---

> ### Author Response · Authors · 2025-11-21
>
> Thank you for your thoughtful review. We have addressed the quoted comments and questions below.
>
> ##  Scale of the study
> > Training new models has the problem that the models are comparetively small and hence one needs to wonder if the findings are transferable
>
> Thank you for raising the question of transferability. While we agree that our 1B-scale models are on the smaller end of large language models, there have been many performant models at this scale, including Qwen-3 0.6B and 1.7B, Llama-3.2 1B and 3B, Gemma-3 1B, and the 270M!, 3B SmolLM models.  Our choice of this scale is also practical---we lack the computational budget to train larger-scale models, especially in the large quantity we consider in our paper. We also note that it is standard practice to reuse the same tokenizer within a model family regardless of the model size. Consequently, the 1B parameter scale offers a tractable, efficient, and highly relevant setting to study the effects of off-the-shelf tokenizers.
> > I would have loved to see the byte latent transformer in the study as well as decoding methods that target byte level decoding such as Phan 2025.
>
> This is an excellent point. We evaluated the BLT-1B model and updated the appendix (Table 19) with the results. While it doesn’t show additional robustness, we refrain from commenting too much on it as it was not trained in the same controlled fashion as the models in TokSuite.
> ## Benchmark
> > Can you justify the choices of the benchmark for me? Do you know if this is in any sense complete?
>
> We agree that the benchmark could be expanded (also raised by 7Sj1) but believe that for the first version of a benchmark that is hand-curated and validated by native speakers, it represents a strong and necessary starting point. Unlike work focusing on randomly generated perturbations, we specifically target real-world scenarios that naturally occur in a multilingual environment.
>
> To this end, the four languages in the study cover different linguistic intricacies :
> - Farsi uses the Arabic script and presents unique challenges where the same text can be represented by optional diacritics, which break tokenization.
> - Mandarin Chinese not only uses a different script but is also an isolating language. We also cover Pinyin and its errors, which is rarely found in the training data but is an essential part of daily communication.
> - Turkish, an agglutinative language with six additional letters in its alphabet, is rich in grammar that severely impacts word form and tokenization.
> - Italian is representative of fusional Latin languages with complex inflectional patterns and accents.
>
> TokSuite includes 40 parallel questions in all five languages that could further be used in studies investigating multilingual knowledge. We select language-specific as well as common perturbations so that the perturbation could be applied to most---all in many cases---canonical questions in that language. Each language has >485 examples (Table 5), which is significant for a hand-curated dataset. We hope that the failure modes observed in our study, and the fact that no single model can handle all categories, serves as a critical first step for the community to pay closer attention to these multilingual robustness issues.
> >  I also wonder about langruages like chinese or thai…. During generation one token may span only part of the semantic meaning aka the opposite of the prompt boundary problem.
>
> This is an excellent point and we agree that it is an important issue for logographic and isolating languages. Our current work partially addresses it for Chinese through intrinsic metrics such as subword fertility and the proportion of continued words (Appendix C), and our text-completion benchmark captures input-side effects of subword fragmentation. However, it does not directly probe generation-time boundary errors, such as producing incomplete morphemes or partial semantic units. A dedicated evaluation of output-side fragmentation in languages like Chinese or Thai, e.g., correlating intrinsic segmentation metrics with generative quality, would be valuable but is beyond the scope and compute constraints of this submission; extending TokSuite to more Asian languages and generation-focused evaluations is a natural direction for future work. Another reason we limit it to these five languages is to make sure that all models can reach a baseline performance in each to be able to make tokenization-related claims.
>
> Additionally, the effect of tokenization on generation is even more difficult to isolate than its effect on the input as we don’t expect the model to generate things like typos which result in easily observed tokenization differences.

---

> > ### Author Response · Authors · 2025-11-21
> >
> > ## BPE
> > > In appendix a can you clarify how all the different bpe tokenizers differ? They all use the same code to my understanding but with different settings what are the settings?
> >
> > We refer the details to Tables 2 and 3 in the Appendix. The BPE-based tokenizers in Appendix A share the same underlying algorithm but differ in the configuration choices made during their original training. These include: (1) vocabulary size (e.g., 32k vs 50k vs 256k); (2) pre-tokenization rules (whitespace handling, digit splitting, contraction splitting, Unicode normalization); (3) merge constraints and continuation-marker schemes (e.g., GPT-2 byte-fallback, WordPiece “##”, SentencePiece “▁”); (4) the training corpus used to learn merges; and (5) normalization and OOV strategies such as byte-fallback vs UNK. We use the tokenizers exactly as released by their respective model families (GPT-2, Phi-3, Llama-3.2, Aya, Gemma-2, etc.), so the differences arise from their original training configurations rather than code differences in our implementation.

---

> > > ### Comment · Reviewer_7UAN · 2025-11-27
> > >
> > > I know that, I am saying please add those details into your paper. Chasing this through the literature takes so much time and your future readers will appreciate this work.

---

> > ### Comment · Reviewer_7UAN · 2025-11-27
> >
> > **Design:** Thank you for your thoughtful response on the design of the study. I came to appreciate these choices more.
> >
> >
> > **Scaling:** Yes, models typically use the same tokenizer across scales, true. However, why would I expect bad downstream performance to prevail, it could be solved by just scaling. I understand that you are computationally restricted, I am not asking for more empirical evaluations, but a better argument as to why your results are relevant at a larger scale too. And
> > I did not see you work that out clearly.
> >
> >
> > **Decoding algorithms:** If you can I would appreciate seeing the results of decoding algorithms that specifically address tokenization issue, after all a mean reviewer could say "Why is there a Benchmark for a problem that has already been solved?"
> > I mean specifically token healing [1,2] and token alignment [3] and CED [4].
> >
> > What I appreciate about [4] is that their algorithm comes with a statistical guarantee: the cross entropy for any string of the tokenized model and the byte level model are the same. Only the decoding algorithm is different. Hence this method can be used for further analysis (training / inference problems).
> >
> > I think if you include at least some of the decoding solutions and show that they can not handle the bench either I am happy to recommend the paper for acceptance. Note that all of them are only decoding schemes, hence are not computaitonally expensive to run and all algorithms have been open sourced.
> >
> >
> > [1] Dagan, Gautier, Gabriel Synnaeve, and Baptiste Roziere. "Getting the most out of your tokenizer for pre-training and domain adaptation." arXiv preprint arXiv:2402.01035 (2024).
> >
> > [2] Roziere, Baptiste, Jonas Gehring, Fabian Gloeckle, Sten Sootla, Itai Gat, Xiaoqing Ellen Tan, Yossi Adi et al. "Code llama: Open foundation models for code." arXiv preprint arXiv:2308.12950 (2023).
> >
> > [3] Athiwaratkun, Ben, Shiqi Wang, Mingyue Shang, Yuchen Tian, Zijian Wang, Sujan Kumar Gonugondla, Sanjay Krishna Gouda et al. "Token alignment via character matching for subword completion." In Findings of the Association for Computational Linguistics: ACL 2024, pp. 15725-15738. 2024.
> >
> > [4] Phan, Buu, Brandon Amos, Itai Gat, Marton Havasi, Matthew Muckley, and Karen Ullrich. "Exact byte-level probabilities from tokenized language models for fim-tasks and model ensembles." arXiv preprint arXiv:2410.09303 (2024).

---

> ### Author Response · Authors · 2025-12-03
>
> We thank the reviewer for their additional comments.
>
> > I know that, I am saying please add those details into your paper. Chasing this through the literature takes so much time and your future readers will appreciate this work.
>
> While we cannot confirm which part of the rebuttal the reviewer was referring to, we updated the introduction to motivate the choice of the languages in TokSuite.
>
> ## Scale
>
> > Scaling: Yes, models typically use the same tokenizer across scales, true. However, why would I expect bad downstream performance to prevail, it could be solved by just scaling. I understand that you are computationally restricted, I am not asking for more empirical evaluations, but a better argument as to why your results are relevant at a larger scale too. And I did not see you work that out clearly.
>
> We agree with the reviewer that it is hard to translate these results to a scaling argument immediately. To address these concerns directly, we trained a **7B model** with the Llama-3.2 tokenizer using the same exact training recipe, duration and data as the 1B model. This means the larger model has seen the exact same raw text in the exact same order as the 1B Llama-3.2 model, differing only in parameter scale.
>
> Notably, we observe very small gains with the 7B model compared to the 1B model. While the canonical performance on the benchmark does improve with scale (see Table X.2), it performs roughly the same in all categories except those related to noise.
>
> Table X.1: Scaling experiment - Comparing 7B and 1B model that is trained with identical tokenizer and training recipe (We added this table to the manuscript as Table 20)
>
> | Model              |   Input (Non-EN) |   Diacritics (Non-EN) |   Orthographic Errors (EN) |   Orthographic Errors (Non-EN) |   Morphological (EN) |   Morphological (Non-EN) |   Noise (EN) |   Noise (Non-EN) |   LaTeX |   STEM (EN) |   Unicode |   Average |
> |:------------------------|-----------------:|----------------------:|---------------------------:|-------------------------------:|---------------------:|-------------------------:|-------------:|-----------------:|--------:|------------:|----------:|----------:|
> | 7B|0.3  |0.52 |0.05 |0.06 |0.24 |0.08 |0.09 |0.14 |    0.17 |0.26 |      0.6  |      0.23 |
> |1B|0.33 |0.55 |0.11 |0.1  |0.25 |0.08 |0.15 |0.24 |    0.18 |0.29 |0.59 |      0.26 |
>
>
> 7B model does perform better in the canonical subset but not in the perturbed subsets.
>
> Table X.2: Canonical performance improves with scale
>
> | Model              |   Chinese Canonical |   English Canonical |   Farsi Canonical |   Italian Canonical |   Math Canonical |   STEM Canonical |   Turkish Canonical |   Average |
> |:------------------------|----------------------------------------------------:|----------------------------------------------------:|--------------------------------------------------:|----------------------------------------------------:|-------------------------------------------------:|-------------------------------------------------:|----------------------------------------------------:|----------:|
> | Llama-3.2-7B |0.9   |1|0.83 |0.98 |1|0.93 |0.93 |  0.94 |
> | Qwen-3|0.7   |0.95  |0.85  |0.98 |0.95 |0.89 |0.85  |  0.88 |
> | Llama-3.2|0.8   |1|0.75  |0.98 |0.81 |0.89 |0.85  |  0.87  |
> | XGLM|0.88 |0.95  |0.8   |0.93 |0.71 |0.89 |0.85  |  0.86 |
> | Gemma-2|0.78 |0.98 |0.75  |0.98 |0.81 |0.89 |0.83 |  0.86 |

---

> > ### Author Response · Authors · 2025-12-03
> >
> > ## Decoding
> >
> > We appreciate the reviewer’s suggestion to situate TokSuite within the literature on decoding algorithms. We emphasize that TokSuite benchmark evaluates the model’s **ability to process a perturbed variation of the input**, while the choices are held constant in majority of examples, unless it is impossible to apply the perturbation to the input. We evaluate each choice as a context-continuation log-likelihood normalized by byte-length.  Since our perturbations occur at the input (context) level and not the output choices, these decoding schemes are not directly comparable in our setting.
> >
> >
> > [1-2] Methods like Token Alignment and Token Healing are primarily designed to address ambiguity or failure modes during output generation and at token boundaries A simple form of token healing is already implemented in the log-likelihood evaluation function of the lm-eval-harness framework we use, see [_encode_pair](https://github.com/EleutherAI/lm-evaluation-harness/blob/b315ef3b05176acc9732bb7fdec116abe1ecc476/lm_eval/api/model.py#L390) function.
> >
> > We therefore prioritized testing the exact byte-level technique from Phan et al.
> >
> > [4] **CED** The CED method proved challenging for comprehensive evaluation:
> > - SentencePiece constraint: The publicly available CED implementation relies heavily on SentencePiece tokenizers; among our models, only Phi-3, Gemma-2, and XGLM use this type. We ran the CED evaluation (labeled Gemma-2-CED) specifically for our Gemma-2 model, which performed the worst among our SentencePiece tokenizers.
> > - Computational cost: Due to byte-level rolling the evaluation takes much longer (a few hours on a subset of 40 examples compared to a few minutes).
> > - Multilingual complexity: The publicly available implementation relies heavily on space-based token delimiters and SentencePiece tokenizers, requiring extensive manual modification to handle non-Latin languages in our benchmark (Chinese and Farsi).
> >
> > While the CED algorithm mitigates the drops in the Gemma-2 model, it still fails to fully solve the issue; the robustness performance only brings it close to the top of Table 1 (TokenMonster). This confirms that TokSuite remains a challenging, unsolved benchmark that warrants further investigation.
> >
> > Table X.3 CED-applied Gemma-2 model
> > | Model           |   Input (Non-EN) |   Diacritics (Non-EN) |   Orthographic Errors (EN) |   Orthographic Errors (Non-EN) |   Morphological (EN) |   Morphological (Non-EN) |   Noise (EN) |   Noise (Non-EN) |   LaTeX |   STEM (EN) |   Unicode |   Average |
> > |:---------------------|-----------------:|----------------------:|---------------------------:|-------------------------------:|---------------------:|-------------------------:|-------------:|-----------------:|--------:|------------:|----------:|----------:|
> > | TokenMonster|0.23 |0.33 |0.09 |0.02 |0.23 |-0.05 |0.11 |0.19 |0.23 |0.11 |0.52 |0.18 |
> > | Gemma-2-CED|0.23 |0.38 |0.17 |0.03 |0.23 |-0.09 |0.16 |0.19 |0.13 |0.15 |0.35 |0.17 |
> > | Gemma-2|0.32 |0.43 |0.14 |0.15 |0.24 |0.03 |0.16 |0.25 |0.22 |0.37 |0.57 |0.26 |
> >
> >
> > Table X.4: Canonical Accuracy of Gemma-2 Model and Its Exact-Byte-Level Correspondent
> >
> > | Model   |   Chinese Canonical |   English Canonical |   Farsi Canonical |   Italian Canonical |   Math Canonical |   STEM Canonical |   Turkish Canonical |   Average |
> > |:-------------|----------------------------------------------------:|----------------------------------------------------:|--------------------------------------------------:|----------------------------------------------------:|-------------------------------------------------:|-------------------------------------------------:|----------------------------------------------------:|----------:|
> > | Gemma-2      |0.78 |0.98 |0.75 |0.98 |0.81 |0.89 |0.83 |  0.86 |
> > | TokenMonster |0.78 |0.88 |0.7  |0.85  |0.67 |0.77 |0.68 |  0.76 |
> > | Gemma-2-CED  |0.68 |0.78 |0.7  |0.68 |0.52  |0.66 |0.58 |  0.65   |

---

### Author Response · Authors · 2025-12-03

We thank all reviewers for providing valuable feedback to our submission and their responses during the rebuttal period. We would like to provide a meta-summary of the clarifications and changes that we made during the rebuttal period.

##  Scope of the TokSuite Benchmark

TokSuite is a hand-curated high-quality multiple-choice completion style benchmark containing linguistically relevant examples in five languages (English, Turkish, Farsi, Italian, and Chinese) with 40 parallel canonical questions translated into each language and perturbed considering the subtleties of each language. TokSuite also contains subsets covering mathematics and STEM.  We clarified the rationale for selecting these languages in the Introduction, demonstrating their representativeness.



## Model Size & Scaling

TokSuite contains 14 language models with 1 billion non-embedding parameters trained identically up-to their tokenizers. To address reviewers questions about the transferability of our results to different parameter scales, we trained a 7B model with the exact same tokenizer, training recipe, data and duration as our Llama-3.2-1B baseline. Notably, we observe very limited difference between the 7B model and the 1B model (see Table 20 in the manuscript). While the canonical performance on the benchmark does improve with scale, it performs roughly the same in all categories except those related to noise.
To further study scaling behavior on our benchmark, we are also training a 300M model to be added to the camera-ready version.

## Statistical Significance and Benchmark Sensitivity
We significantly enhanced the statistical rigor of the paper to address concerns raised by reviewers (7UAN, RRpy, 7Sj1) and expanded it in Appendix F.1.

- Bootstrap and Variance: We estimate distributional statistics through a 10,000-trial bootstrap, confirming that all discussed performance differences are statistically significant (larger than one standard deviation) and adding these details to the Appendix (Table 21).

- Categorical Sensitivity: We performed the Paired Wilcoxon Signed-Rank Test to confirm that every perturbation category (except Unicode) shows a statistically significant differentiation between the best and worst-performing models, proving the benchmark's high sensitivity. This disaggregation is visualized in Figure 6, showing that no single tokenizer excels universally.

### Advanced Decoding and Unsolved Problems

As per reviewer 7UAN’s suggestion, we situated our work better in the literature around advanced decoding algorithms. We tested the CED algorithm on our Gemma-2 model. While we observe improvements on the Gemma-2 model, the CED method was only able to bring its robustness score close to the performance of the best tokenizer (TokenMonster), validating TokSuite remains a challenging, unsolved benchmark for language models.

## Training Procedure

### Token Budget vs. Text Budget

We clarified the training procedure, the deterministic nature of our dataloader, and the resulting amount of raw text (bytes) consumed by each TokSuite model. As an alternative experiment to address fairness concerns, we are currently training 3 models (Comma, Gemma-2, Qwen-3) with a controlled text budget (same total bytes seen). Since the Comma model requires training for $\sim180$ Billion tokens to match the byte budget, this experiment will be added to the final camera-ready version once complete.

---

### Meta-Review · Area_Chair_KqFg · 2026-01-07

**Summary:**

Reviewers raised concerns about the extent to which TokSuite provides genuinely new scientific insights beyond confirming previously observed tokenizer-related phenomena under a more controlled setup. While the experimental design is careful, reviewers questioned whether the proposed benchmark and findings substantially advance understanding of tokenizer effects on downstream model behavior, or primarily serve as a large-scale empirical validation. Additional concerns focused on the limited diversity of model architectures and training regimes, as well as the unclear implications for practical model design decisions.

**Reviewer Concerns:**

The rebuttal successfully clarified the motivation behind TokSuite and better articulated the authors’ intent to provide an analysis framework rather than a novel modeling technique, partially addressing concerns about positioning and scope. Some questions regarding experimental controls and benchmark construction were also adequately explained. However, central concerns about the level of novelty, the incremental nature of the empirical findings, and the broader impact on tokenizer research remain largely unresolved. In particular, the rebuttal did not convincingly demonstrate that the insights enabled by TokSuite would be difficult to obtain through existing evaluation practices.

**Reviewer Scores:**

Had reviewers been able to participate fully in the post-rebuttal discussion, it is likely that scores would have remained largely unchanged, with at most minor upward adjustments reflecting improved clarity rather than increased enthusiasm. The rebuttal improved understanding of the work but was unlikely to shift reviewers’ overall assessment of novelty and impact.

---

### Decision · Program_Chairs · 2026-01-26

Reject